# Novel LDPE/Chitosan Rosemary and Melissa Extract Nanostructured Active Packaging Films

**DOI:** 10.3390/nano9081105

**Published:** 2019-08-01

**Authors:** Aris Giannakas, Constantinos Salmas, Areti Leontiou, Dimitrios Tsimogiannis, Antigoni Oreopoulou, Joerg Braouhli

**Affiliations:** 1Laboratory of Food Technology, Department of Business Administration of Food and Agricultural Enterprises, University of Patras, GR-30100 G Agrinio, Greece; 2Department of Materials Science & Engineering, School of Engineering, University of Ioannina, GR-45110 Ioannina, Greece; 3Department of Chemical Engineering, Laboratory of Food Chemistry and Technology, National Technical University of Athens, 5 Iroon Polytechniou, 15780 Zografou, Athens, Greece; 4VIORYL, Chemical & Agricultural Industry, Research S.A., 28th km National Road Athens-Lamia, 190-14 Afidnes, Greece; 5Antir S.A., GR-30100 G Megali Khora Agrinio, Greece

**Keywords:** LDPE, chitosan, rosemary extract, Melissa extract, active packaging films, chitosan/extract hybrids, LDPE/chitosan/extract nanostructures

## Abstract

The increased global market trend for food packaging is imposing new improved methods for the extension of shelf-life and quality of food products. Active packaging, which is based on the incorporation of additives into packaging materials, is becoming significant for this purpose. In this work, nanostructured low-density polyethylene (LDPE) was combined with chitosan (CS) to aim for a food packaging development with an increased oxygen permeability barrier and higher antimicrobial activity. Furthermore, essential oil extracts as rosemary (RO) and Melissa (MO) were added to this packaging matrix in order to improve its antioxidant properties and vanish food odor problems. The novel nanostructured active packaging film was tested using laboratory instrumental methods, such as thermogravimetry (TG), Fourier-transform infrared (FTIR) spectrometry, the X-ray diffraction (XRD) method, a dilatometer for tensile properties (DMA), and an oxygen permeation analyzer (OPA). Moreover, laboratorian tests according to ASTM standards were carried out for the estimation of water sorption, water vapor permeability, overall migration, and, finally, the antioxidant properties of such films. The experimental results have indicated that the final material exhibits advanced properties. More specifically, chitosan addition was observed to lead to an enhanced oxygen and water-vapor permeability barrier while the extracted essential oil addition led to enhanced tensile strength and antioxidant properties.

## 1. Introduction

Nowadays, there is an increasing global trend aimed at enhancing the cyclic economy, the real economy, and the quality of products. For this purpose, during the product design stage there is now a global effort to incorporate raw materials such as byproducts, biomass, and/or biowastes which have zero or negative added value and which have been biodegradable, with a zero environmental fingerprint. Manufacturers also exploit new scientific methods like nanotechnology in order to extend the shelf-life and quality of products. Especially in the case of the food industry, what is also mostly preferable for consumers is the use of biological instead of chemical preservatives, e.g., essential oil extracts as antioxidants. In recent years, a lot of research effort has been oriented to reducing environmental pollution from non-biodegradable synthetic polymers which are used as food packaging materials. For this purpose, composite packaging films have been produced by mixing polymers and biopolymers. In this work the basic concept is to combine the advanced properties of three materials in order to produce an improved film for use as food packaging.

The first material is low-density polyethylene (LDPE), which is a low-cost material exhibiting good processability, zero odor, zero toxicity, low water vapor permeability, and high heat transfer resistance [1]. It also exhibits high toughness and flexibility even at low temperature conditions [2]. Thus, LDPE is one of the most widely used polymers for flexible active packaging films [3].

The second material is chitosan, CS, which is a linear polysaccharide made by treating a food byproduct, i.e., the chitin shells of crustaceans. CS films have great potential to be used as packaging material [4] due to their biodegradability, nontoxicity, antioxidant, and antimicrobial activity [5]. It is also a very good biopesticide. 

The third material is extracted oil from aromatic plants, which is a great source of active compounds suitable for use in active packaging films [6,7]. The active compounds which are included in the extract of aromatic plants are phenolic acids and flavonoids, which have antioxidant and antimicrobial properties [8].

The problem with CS usage is that although CS can produce perfect films via a solution casting method [9] it cannot be blended in industrial extruders. Moreover, CS’s hydrophilicity makes it not compatible with the most synthetic polymers. A method to overcome this problem is to graft the polymers’ surfaces with compatibilizers such as maleic anhydride [2,10,11,12,13,14,15]. The modification of LDPE films with CS could produce new high added value products. There are many works within the literature in which LDPE/CS films were developed as novel packaging films [2,12,16,17,18,19,20]. Park et al. in 2010 incorporated CS into LDPE to create active packaging films and to extend the shelf-life of sliced red meat. Wang et al. in 2015 developed Linear LDPE (LLDPE)/CS blown films with good mechanical properties and barrier performance using LLDPE-grafted-maleic anhydride (LLDPE-g-MAH) as a compatibilizer and CS content of up to 20% *w*/*w* Reesha et al. in 2015 [12] developed an antimicrobial packaging film by homogeneous embedding of 1, 3, and 5% *w*/*w* of CS in an LDPE matrix using maleic-anhydride-grafted LDPE as a compatible agent. In this work, the analysis of storage quality indices revealed extension of shelf-life for Tilapia packaged with novel composite films which had CS incorporated into them compared to the shelf-life of Tilapia packaged with virgin LDPE film.

According to the literature [21,22,23], rosemary (RO) extracts have been incorporated within biopolymer-based active packaging films such as starch [21], whey protein [22], and furcellaran/gelatin [23]. As far as we know, there have been no reports that Melissa (MO) extracts have been used for active packaging applications.

In this work, novel bioactive LDPE/CS films were developed in which CS was modified with RO and MO water/ethanol extracts prior to being incorporated into an LDPE matrix. Extrusion molding was used as a preparation method and polyethylene grafted with co-maleic anhydride (PEGMA) was used as a compatibilizer. Pure CS, CS modified with RO extract (CS_RO), and CS modified with MO extract (CS_MO) were characterized using XRD, thermogravimetry (TG), and FTIR instruments. LDPE/CS, LDPE/CS_RO, and LDPE/CS_MO films were morphologically and structurally characterized by XRD and FTIR measurements. Packaging performance evaluation was carried out via measurements on tensile properties, water and oxygen barrier properties, water sorption, the overall migration rate, and antioxidant activity tests for all novel LDPE/CS_RO and LDPE/CS_MO films, and these measurements were compared to similar ones for LDPE/CS films. 

## 2. Materials and Methods

### 2.1. Materials

LDPE was supplied by Aldrich (cat. no. 428027), with melt index = 1.5 g/10 min (190 °C/2.16 kg) and d = 0.922 g cm^−3^. Polyethylene grafted with co-maleic anhydride (PEGMA) was supplied by Aldrich (cat. no. 426946), with melt index = 1.70 g/10 min, Tg = 120 °C, and d = 1.27 g cm^−3^. CS with medium molecular weight, a viscosity of 200–800 cP, 1% *w*/*w* in 1% acetic acid at 25 °C, and deacetylation degree 75–85% was supplied from Sigma–Aldrich (cat. no. 448877). RO and MO extractions were offered by ANTHIR S.A. For the preparation of RO and MO extraction a stim distillation process was carried out. The solvent used for the extraction was a mixture of water and ethanol 1:1 v/v. Results from this HPLC–DAD method, which was proposed by Tsimogiannis et al. [24], showed an RO extract composition as follows: 213.7 mg/L rosmarinic acid, 8.4 mg/L rest phenolic acids, and 598.9 mg/L total flavonoids (flavone-flavonol glycosides). The same method resulted in an MO extract composition as follows: 1913.1 mg/L rosmarinic acid, 404.7 mg/L rest phenolic acids, and 116.1 mg/L total flavonoids (flavone-flavonol glycosides).

### 2.2. Preparation of Active Films

#### 2.2.1. Preparation of CS_RO and CS_MO Hybrids

5 g of CS was weighed and added to 100 mL of RO and MO extract in a glass beaker. The mixture was stirred for 24 h and then the solvent was evaporated. The obtained CS_RO and CS_MO hybrids (Scheme 1) were dried in an oven at 120 °C for 24 h and were stored for further use and characterization. 

#### 2.2.2. Preparation of Nanostructured Films

LDPE/CS blends were prepared in a lab scale, twin-screw extruder (Haake Mini Lab II, ThermoScientific, NTISEL, S.A., Athens, Greece). The CS, CS_RO, and CS_MO content was fixed at 17.6, 26.4, and 35.2% *w*/*w*., respectively, according to a previous work [25]. PEGMA was used as a compatibilizer. The first blending step was carried out at 30 min blending time at 140 °C and a rotor speed of 25 rpm. The second blending step took place at 30 min blending time at 140 °C and a rotor speed of 50 rpm. The final step was completed in 30 min blending time at 140 °C and a rotor speed of 100 rpm. In Table 1 the used amounts of LDPE, PEGMA, CS, CS_RO, and CS_MO are listed, as well as the adopted code names of all composites in this work. The blends obtained from the lab scale twin-screw extruder were hot pressed into films for 5 min at 110 °C under 2 MPa constant pressure using a hydraulic press with heated platens. 

### 2.3. XRD Analysis

The morphological evaluation of the CS_RO and CS_MO hybrids and pure CS, LDPE/CS, LDPE/CS_RO, and LDPE/CS_MO nanostructured films were estimated from the XRD pattern obtained using a Brüker D8 Advance X-ray diffractometer (Bruker, Analytical Instruments, S.A. Athens, Greece) equipped with a LINXEYE XE High-Resolution Energy-Dispersive detector. Typical scanning parameters were set as follows: two theta range 2–40° for powder samples and 2–30° for film samples; increment 0.03°; PSD 0.764. 

### 2.4. FTIR Spectrometry

The chemical structures of the modified CS_RO, CS_MO, and raw CS powder samples as well as of the obtained LDPE/CS, LDPE/CS_RO, and LDPE/CS_MO nanostructured films were confirmed by IR spectra measurements. Infrared (FTIR) spectra, which were the average of 32 scans at 2 cm^−1^ resolution, were measured with an FT/IR-6000 JASCO Fourier transform spectrometer (JASCO, Interlab, S.A., Athens, Greece) in the frequency range 4000–400 cm^−1^.

### 2.5. Thermogravimetric/Differential Thermal Analysis

Thermogravimetric (TGA) and differential thermal analysis (DTA) were performed on modified CS_RO and CS_MO hybrids and raw CS powder samples using a Perkin-Elmer Pyris Diamond TGA/DTA instrument (Interlab, S.A., Athens, Greece). Samples of approximately 5 mg were heated under an N_2_ flow from 25 to 700 °C at a rate of 5 °C/min. These measurements were carried out for the calculation of RO and MO amounts adsorbed in CS_RO and CS_MO hybrids.

### 2.6. Tensile Properties

Tensile measurements were carried out on all prepared nanostructured films according to the ASTM D638 method using a Simantzü AX-G 5kNt instrument (Simantzu. Asteriadis, S.A., Athens, Greece) Three to five samples of each film were tensioned at an across head speed of 2 mm/min. The samples were dumb-bell shaped with gauge dimensions of 10 mm × 3 mm × 0.22 mm. Force (N) and deformation (mm) were recorded during the test. Based on these data and the gauge dimensions, stress, stain, and modulus of elasticity were also calculated. 

### 2.7. Water Sorption

Selected films were cut into small pieces (12 mm × 12 mm), desiccated overnight under vacuum, and weighed to determine their dry mass. The weighed films were placed in closed beakers containing 30 mL of water (pH = 7) and stored at T = 25 °C. The sorption plots were evaluated by periodical weighting of the samples until equilibrium was reached and according to the equation
W.G. (%) = (m_Wet_ − m_Dry_)/m_Dry_ × 100(1)
where m_Wet_ and m_Dry_ are the weight of the wet and dry film respectively, where W.G. is the Water Gain.

### 2.8. Water Vapor Permeability (WVP)

Water vapor permeability of all nanostructured films was determined at 38 °C and 50% RH according to the ASTM E96/E 96M-05 method using a handmade apparatus and following the methodology described extensively in our previous publications [9,26].

### 2.9. Oxygen Permeability (OP)

The oxygen transition rate (OTR) was analyzed using an oxygen permeation analyzer (8001, Systech Illinois Instruments Co., Johnsburg, IL, USA). The tested samples were evaluated at 23 °C and 0% RH according to the ASTM D 3985 method. OTR values were measured in cc O_2_/m^2^/day. The OP values of the tested samples were calculated by multiplying the OTR values with the average film thickness, which was approximately 350–400 μm. The mean OTR value for each kind of film resulted from the measurements of three samples.

### 2.10. Overall Migration Test

The overall migration measurements of different LDPE/CS films were carried out according to the USFDA 176:170 test procedure [27]. The film pouches were filled with 250 mL stimulating solvent (water) at 49 °C for 24 h. After exposure to the atmosphere for a specified duration the film was dried, and the solvent was evaporated. The residues were weighed, and the overall migration residue (OMR) values were calculated as follows: OMR in mg/L = (mass of residue (mg) × 1000)/(Volume of stimulant (mL))(2)

### 2.11. Antioxidant Activity

RO and MO extract antioxidant activity was examined with the DPPH radical method as follows. Four mL of a 70 ppm DPPH–ethanolic solution was mixed with 120 μL of the tested extracts. The resulted solution was vigorously mixed and incubated in a dark place under ambient temperature for 30 min. Sequentially, the absorbance of the liquid sample at 517 nm was tested using a Jasco V-530 photometer. The % antioxidant activity of extracts was calculated using Equation (3), as follows: % Antioxidant activity = (Abs_control_ − Abs_sample_)/Abs_control_) × 100(3)

The DPPH free of extract solution was used to develop the baseline of the instrument.

Antioxidant activity of films was evaluated using 300 mg of small pieces (approximately 3 mm × 3 mm) of each film. The sample was placed in a dark colored glass bottle with a plastic screw cap and filled with 10 mL of 30 ppm (mg/L) ethanolic DPPH solution. After incubation at 25 °C for 24 h in darkness the % antioxidant activity of the films was calculated according to Equation (3).

## 3. Results

### 3.1. XRD

Figure 1a shows XRD plots of pure CS, CS_RO, and CS_MO powders measured in the range of 2–30°. For the raw CS samples two broad peaks at 2θ = 9.4° and at 2θ = 20.3° were observed, which agrees with literature reports [28,29].The peaks correspond to a hydrated crystalline structure and an amorphous structure of CS, respectively [29]. The CS_RO and CS_MO hybrid XRD plots show a small shift of the hydrated CS peak from 2θ=9.4° to higher angles compared to the plot of pure CS. In the same plots no significant position changing of the amorphous CS peak at 20.3° is observed. Thus, modification of CS with both RO and MO extraction does not affect significantly the crystal structure of CS. 

XRD plots of the LDPE/CS, LDPE/CS_RO, and LDPE/CS_MO films are presented in Figure 1b, Figure 1c, and Figure 1d, respectively. In all cases it is obvious that as CS, CS_RO, and CS_MO increase, the characteristic CS peaks at around 2θ = 9.4° and 20.3° become more intensive. At the same time as CS, CS_RO, and CS_MO increase, the characteristic LDPE peaks at around 21.8° and 24.0° shift to smaller angles. These simultaneous observations indicate the effective blending of LDPE chains with CS, CS_RO, and CS_MO chains.

### 3.2. TG Results

Typical TG plots for CS, CS_RO, and CS_MO samples are shown in Figure 2, where two weight loss areas are observed for all samples. An initial weight loss is seen to occur for the temperature range 30–220 °C and this is due to the elimination of the adsorbed moisture by the polysaccharide. For the CS_RO sample this initial weight loss area is extended up to 245 °C, which shows that a higher amount of water was adsorbed. The second stage of weight loss can be observed to be in the temperature range 220–550 °C and is assigned to the decomposition of CS chains [30]. As is shown in Figure 2, the 50% weight loss temperature is higher for both the CS_RO (371 °C) and CS_MO (334 °C) hybrid samples than for the pure CS sample (321 °C). This indicates that the modification of CS with RO and MO leads to a thermal stability enhancement of obtained CS_RO and CS_MO powders. The highest thermal stability obtained was that of the CS_RO sample.

### 3.3. FTIR Results

The FTIR spectra of pure CS as well of the CS_RO and CS_MO hybrids are shown in Figure 3. In accordance with previous reports [12,17,18,25,31] these CS spectra are shown to present three main areas: (i) a broad asymmetric band between 3400 and 2500 cm^−1^ encompassing the CH stretching modes at around 2900 and 2880 cm^−1^ and the overlapped OH and NH stretching vibrations at higher wavenumbers (approximately 3400 cm^−1^); (ii) an area between 1700 and 1200 cm^−1^ which is characteristic of the amide groups; (iii) a strong absorption area between 1200 and 800 cm^−1^ which is characteristic of the CS saccharide structure.

Characteristic peaks of amide I, amide II, and amide III are located at 1650 cm^−1^, 1590 cm^−1^, and 1317 cm^−1^. Other characteristic bands of CS are evidenced at 1161 cm^−1^ and 1051 cm^−1^. The first peak can be attributed to the beta glyosidic bond between carbon 1 and carbon 4 of the CS and the second peak may be associated with the COC stretching of the glucopyranoside ring. Finally, peaks at 1420 cm^−1^ and 1380 cm^−1^ represent the deformation bands of CH_2_ and CH_3_.

After the incorporation of RO and MO into CS (Figure 2, line (2) and line (3)) no additional peaks and no significant wavelength shift occurred, which shows the absence of covalent bonds between RO, MO, and CS [32]. The absence of covalent bonds is beneficial for the controlled release of antioxidant RO and MO extracts from the bioactive films [33]. A significant reduction of the CS_RO spectra and a small increase in the CS_MO spectra is observed compared to the CS spectra. Reduced OH and NH vibration peaks (3400 cm^−1^, 2900 cm^−1^, and 1650 cm^−1^) can be assigned to the reduced stretching of -NH and/or -OH due to the binding interactions between RO and CS. This fact also indicates a hydrogen bonding formation between RO components and CS chains [34]. Thus, for both RO and MO extracts a physical adsorption into CS chains is evidenced. This physical adsorption is stronger via hydrogen bonding for RO extracts. This hypothesis is consistent with TG results which are discussed above, where a higher increase of thermal stability in the case of the CS_RO sample was indicated.

As is shown in Figure 4 and in accordance with previous works [2,12,16,17,18,25], all LDPE/CS, LDPE/CS_RO, and LDPE/CS_MO film FTIR plots include the characteristic peaks of LDPE. The –CH_3_ asymmetric stretching, –CH_2_ wagging, and -CH_2_ rocking in particular are depicted by the peaks at 1460 and 715 cm^−1^, while the –CH_2_ symmetric stretching peaks are at 2913 and 2844 cm^−1^. For all films, the characteristic peak of the epoxy functional group of PEGMA, which usually occurs in the range 925–899 cm^−1^, was not detected. This fact indicates that during blending the epoxy group was cut off by interacting with the hydroxyl and amide groups of CS. In all FTIR plots the characteristic peaks of CS are obvious. As has been mentioned above, these peaks lie in the range of 1900–1400 cm^−1^ and 3800–3200 cm^−1^. It is also evidenced that as the CS content increased, the detected LDPE bands decreased and the CS bands were enhanced. This result is consisted with previous reports [17] and it indicates the effective blending of LDPE with pure CS, CS_RO, and CS_MO. As is shown by the dotted tetragonal shape in Figure 4, spectra of all LDPE/CS composite films show a shift in characteristic peaks of some bands (amino and carbonyl groups present in the CS) within the range 1719–716 cm^−1^. According to the literature [2], the interactions between chemical groups on dissimilar polymers should theoretically cause a position shift of peaks of the participating groups. In the present work this kind of behavior is observed for certain peaks. The shift of peaks is a clear indicator of the interaction between CS chains and the LDPE matrix.

Moreover, the comparative analysis of all LDPE/CS_RO and LDPE/CS_MO FT-IR plots in Figure 4 leads to the conclusion that the characteristic CS peaks are more intense compared to these of the respective LDPE/CS films (Figure 3, 1900–1400 cm^−1^ and 3800–3200 cm^−1^). This indicates that the modification of the CS with RO and MO enhances its blending with the LDPE chains.

### 3.4. Tensile Properties

Typical strain-stress curves of all tested films are presented in Figure 5. The average values, the standard deviation of Young’s modulus (E), the tensile strength (σ_uts_), and the elongation at break (ε_b_) were calculated based on Figure 5 curves and are tabulated in Table 2.

In Figure 5a what is presented is the % variation of Young’s modulus (E) and the elongation at break (ε_b_) of all tested films compared to corresponding values of the pure LDPE film.

As is shown in Figure 5, the stress values of all LDPE/CS blends increased as the CS loading also increased. The strain values reduced as rigid CS particles were included in blends. CS addition in raw LDPE is shown to decrease the elongation of extruded films (Table 2), which is in accordance with literature [17]. It is also obvious from Table 2 that for all LDPE/CS, LDPE/CS_RO, and LDPE/CS_MO films, the tensile strength values (σ_uts_ values in Table 2) show a decreasing trend as the CS content increases. Because CS is a brittle material, an increase of CS content results in a decrease of ductility [13]. On the contrary, Young’s modulus (E) values (E values in Table 2) show an increasing trend with large CS content. This is a typical behavior for thermoplastic materials blended with brittle materials such as CS [13,16]. A further increase in E values is observed for the LDPE/CS_RO and LPDE/CS_MO films while the largest increase in E values is observed for the LDPE/CS_MO films. E values for LDPE/CS2, LDPE/CS3, LDPE/CS_RO2, LDPE/CS_RO3, LDPE/CS_MO1, LDPE/CS_MO2, and LDPE/CS_MO3 films were found to be approximately 6.9%, 39%, 16%, 43%, 5.2%, 14%, and 48% respectively higher than the values of the pure LDPE film (Figure 5). 

Modification of CS with RO and MO was found to lead to the development of LDPE/CS_RO and LDPE/MO nanostructured films, which exhibit higher stiffness than LDPE/CS composite films. This result is consistent with FTIR results, where it was shown that the modification of CS with RO and MO improves the blending of modified CS_RO and CS_MO with the LDPE matrix.

### 3.5. Water Barrier–Water Sorption

Calculated WVP values as well as water sorption values are presented in Table 3. In Figure 6 the comparison of the % variation of WVP values for all tested films is depicted alongside the values of the pure LDPE film.

Both W.V.P. and water sorption values show the same trend (Table 3). This was expected as both properties are affected by the hydrophilicity of tested films. PE films are known to be highly hydrophobic and relatively not very permeable to water vapor. The WVP value for the pure LDPE film was found to be 17.7 g/m^2^/day. As was expected, no water sorption was detected for the pure LDPE film. All films were found to exhibit higher WVP and water sorption values than the pure LDPE film. WVP and water sorption values were observed to increase with increasing CS content. This result is in accordance with previous reports [2,12,25] where it has been mentioned that the hygroscopic CS layer acts as a water reservoir on the LDPE surface. This promotes its WVP value significantly. Modification of CS with RO and MO increases the hydrophobicity of developed LDPE/CS_RO and LDPE/CS_MO composites. WVP and water sorption values were found to be decreased for the LDPE/CS_RO films and further decreased for the LDPE/CS_MO films. This result is consistent with TG results where higher amounts of adsorbed water were measured for the CS_RO sample than the CS_MO sample. The lowest WVP values were obtained for the films with the lowest CS, CS_RO, and CS_MO content. Thus, the WVP values of the LDPE/CS1, LDPE/CS_RO1, and LDPE/CS_MO1 films were found to be 53%, 27%, and ~0% higher than the corresponding value of the pure LDPE film (Figure 6).

### 3.6. Oxygen Permeability

The pure LDPE film was found to show a higher OP value compared to all the LDPE/CS, LDPE/CS_RO, and LDPE/CS_MO films (Table 3). The OP value of the pure LDPE film was found to be around 185.5 mL/m^2^/day, which is similar to values reported in the literature [18]. Significant differences in OP values were found when CS, CS_RO, and CS_MO were incorporated into LDPE (Table 3). OP values for all LDPE/CS, LDPE/CS_RO, and LDPE/CS_MO films were found to decrease as the CS content increased. This observation is consistent with previous reports [12,18,19]. CS exhibits excellent oxygen barrier properties due to polar interactions in its structure. Much more tortuous paths were developed by the well-dispersed CS, CS_RO, and CS_MO phases in the polymer matrix. This fact enhances the oxygen barrier performance of films [19]. In Figure 6 the % variation of OP values of all tested films is presented compared to the respective OP value of the pure LDPE film. LDPE/CS_RO and LDPE/CS_MO films showed much lower OP values compared to the values of the LDPE/CS films. Thus, the OP value decreased to 35% for the LDPE/CS3 film, to 53% for the LDPE/CS_RO3 film, and to 59% for the LDPE/CS_MO3 film (Figure 6).

### 3.7. Overall Migration Rate

An Overall Migration Rate (OMR) of all LDPE-based tested films was found within the stipulated upper limit of 60 mg/L (Table 3). Pure LDPE exhibited the lowest value of 12.44 mg/L. By increasing CS concentration, the values of migration of all tested films were increased. No significant differences for OMR values were observed among the LDPE/CS, LDPE/CS_RO, and LDPE/CS_MO films. Considering the low migration rate of all these films, it could be concluded that such films can be suitably used for food packaging. 

### 3.8. Antioxidant Activity

Antioxidant activity for RO and MO extracts was determined to be 63.3 ± 2.3% and 37.2 ± 3.1%, respectively, and it was proportional to total flavonoid composition, which was shown to be 598.9 mg/L for RO and 116.1 mg/L for MO extract. The number of flavonoids in such extracts plays a significant role in their antioxidant capacity [35]. 

The obtained antioxidant activity values of all LDPE/CS, LDPE/CS_RO, and LDPE/CS_MO samples are listed in Table 3.

Antioxidant activity values of LDPE/CS active films were found to increase when increasing the CS content and range at 27.1% for LDPE/CS1, 33.9% for LDPE/CS2, and 39.5% for the LDPE/CS3 sample. It is known [36] that CS inhibits reactive oxygen species and prevents the lipid oxidation of food. The average antioxidant activity value of LDPE/CS_RO nanostructured films was found to be approximately 10% higher than the respective antioxidant activity values of the LDPE/CS films. For the LDPE/CS_MO nanostructured films the average antioxidant activity value was observed to be approximately 5% higher than the respective antioxidant activity values of the LDPE/CS films. Obtaining higher antioxidant activity values of the LDPE/CS_RO active films compared with LDPE/CS_MO active films is in accordance with higher antioxidant activity values of RO extract compared to MO extract.

## 4. Conclusions

The main conclusion of this work is that the incorporation of chitosan and essential oils from rosemary and Melissa aromatic plants in a low-density polyethylene matrix is able to produce active packaging films with improved properties with regard to the oxygen and water-vapor permeability barrier, water sorption, tensile strength, antioxidant and antimicrobial activity, and, finally, odor elimination activity. All laboratorian measurements indicated that rosemary and Melissa extract were adsorbed physically on chitosan chains. This modified system improves blending with the LDPE matrix and gives a nanostructured material which is a promising candidate material for the development of an advanced active packaging film.

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
