# Peer review of "Novel LDPE/Chitosan Rosemary and Melissa Extract Nanostructured Active Packaging Films"

_nanomaterials, 2019, doi:10.3390/nano9081105_

Round 1

Reviewer 1 Report

The authors have developed nanostructured LDPE/Chitosan Rosemary and Melissa extracts active packaging films via an extrusion molding process. TG, FTIR and XRD have been applied to characterize these packaging films. The properties of stiffness, ductility, water vapor permeability, water sorption, oxygen permeability and overall migration of packaging films have been also investigated to demonstrate their advantages. This work can inspire the applications of packaging films in the research fields of nanomaterials. Therefore, I would like to recommend this work to publish in Nanomaterials after major revision. The following issues should be addressed to further improve the quality of this work.

1. The resolution of Figure 1 is too low to identify. Please provide high resolution image of Figure 1. Especially, there are some strange slashes in Figure 1a. Please correct.

2. What are the numbers in Figure 2? Please describe in the Caption of Figure 2 or revised manuscript.

3. The resolution of Figure 3 and Figure 4 are also too low. Please provide high quality FTIR spectra.

4. Please increase the quality of all Figures.

Author Response

Dear Editor

First of all we would like to thank Reviewers for their positive comments on our work and for their effort to provide to us valuable suggestions which helped us to improve the presentation and the interpretation of our work.

In general, we would like to note the following changes that made in our manuscript:

1.      The Abstract was fully rewritten.

2.      The Introduction was changed substantially. An extended paragraph was added in the beginning to describe the problem and the idea more clearly. Sequentially, some paragraphs were rearranged and some text was changed.

3.      English language was improved by a native English-speaking colleague.

4.      The resolution of the figures was upgraded as it was suggested by both reviewers.

5.      The conclusion section was fully rewritten.

More specifically:

Reviewer 1

The authors have developed nanostructured LDPE/Chitosan Rosemary and Melissa extracts active packaging films via an extrusion molding process. TG, FTIR and XRD have been applied to characterize these packaging films. The properties of stiffness, ductility, water vapor permeability, water sorption, oxygen permeability and overall migration of packaging films have been also investigated to demonstrate their advantages. This work can inspire the applications of packaging films in the research fields of nanomaterials. Therefore, I would like to recommend this work to publish in Nanomaterials after major revision. The following issues should be addressed to further improve the quality of this work.

 1. The resolution of Figure 1 is too low to identify. Please provide high resolution image of Figure 1. Especially, there are some strange slashes in Figure 1a. Please correct.

Ans.: Improved with higher resolution figure and the slashes have been removed from Figure 1a.

2. What are the numbers in Figure 2? Please describe in the Caption of Figure 2 or revised manuscript.

Ans.: In the revised manuscript, the Figure 2 was revised in a higher resolution figure and the text (lines 199-200) was also revised to describe the temperature numbers in Figure 2.

3. The resolution of Figure 3 and Figure 4 are also too low. Please provide high quality FTIR spectra.

Ans.: The figures were improved with higher resolution.

4. Please increase the quality of all Figures.

Ans.: The figures were improved with higher resolution.

Reviewer 2

1-                  General comment: the article required substantial English editing.

Ans.: English language was improved by a native English-speaking colleague.

2-                  The Abstract: is poorly written. Too many acronyms make reading this abstract difficult. There are multiple terms, such as FTIR, XRD, and TG that are not defined. If terms are used once or twice within the Abstract, there is no need for abbreviation. This could help to lower the number of acronyms in the text and make it more fluent. Some sentenses are in past and some are at present tense (e.g., "Stiffness increases and ductility decreased..."). This needs to be consistent. While there are so many technical reports and numbers squeezed into the Abstract, there seems to be no logical flow/glue to connect these findings together. There is no intro on the significance and reasoning of doing this study. Why these specific reagents used? why each assay was conducted? And at the end, what are the impacts / take home messages of the study?

Ans.: The Abstract was fully rewritten. The terms FTIR, XRD, TG were defined in the abstract. Nevertheless, because of the frequent use of the terms Low-Density PolyEthylene, Chitosan, Rosemary, Melissa, Fourrier Transformed InfraRed, X-Ray Diffraction, ThermoGravimetry in the main body of the manuscript the authors believe that the acronyms are very helpful and they reduce the text length. In any case if reviewers and the editor believe that this should be changed and the acronyms should be skipped then the authors are available to make this changes. The tense of the sentences was corrected. The squeezed numbers of the abstract were vanished. The Introduction now makes the scope of this research clearer. Finally, the impact of this study is now obvious at the end of the Abstract and at the new Conclusion paragraph. The choice of the specific reagents used is now established (lines 40-47, 52-63). The methods that the authors followed are common methods for the characterization of such materials.

3-                   Figures 1, 3, 4, and 5: is not readable. Too small and low quality.

Ans.: The figures were improved with higher resolution.

4-                  The major shortcoming of this work is vague, and incomplete structure of the manuscript. The reasoning and logic as why the study was designed and conducted the way described are not clear. Also, reading through the Conclusion section, it is not clear what the main outcomes of the study were?! 

Ans.: Authors believe that the substantial changes on Abstract, Conclusion and the main text now provide a clear picture of the problem which inspires this study.

Thank you for your considering on our work

Sincerely

Dr. Aris Giannakas

Dr. Constantinos Salmas

Reviewer 2 Report

1- General comment: the article required substantial English editing. 2- The Abstract: is poorly written. Too many acronyms make reading this abstract really difficult. There are multiple terms, such as FTIR, XRD, and TG that are not defined. If terms are used once or twice within the Abstract, there is no need for abbreviation. This could help to lower the number of acronyms in the text and make it more fluent. Some sentenses are in past and some are at present tense (e.g., "Stiffness increases and ductility decreased..."). This needs to be consistent. While there are so many technical reports and numbers squeezed into the Abstract, there seems to be no logical flow/glue to connect these findings together. There is no intro on the significance and reasoning of doing this study. Why these specific reagents used? why each assay was conducted? And at the end, what are the impacts / take home messages of the study? 3- Figures 1, 3, 4, and 5: is not readable. Too small and low quality. 4- The major shortcoming of this work is vague, and incomplete structure of the manuscript. The reasoning and logic as why the study was designed and conducted the way described are not clear. Also, reading through the Conclusion section, it is not clear what the main outcomes of the study were?!

Author Response

(The authors gave the same response as above.)

Round 2

Reviewer 1 Report

The authors have addressed all my concerns. Therefore, I would like to recommend to publish this paper as its current form.

Reviewer 2 Report

The authors have addressed most of the major comments raised by the reviewers. I believe this manuscript now merits publication in Nanomaterials.